# Stem Cells in Clinical Trials on Neurological Disorders: Trends in Stem Cells Origins, Indications, and Status of the Clinical Trials

**DOI:** 10.3390/ijms231911453

**Published:** 2022-09-28

**Authors:** Eugenia D. Namiot, Jenni Viivi Linnea Niemi, Vladimir N. Chubarev, Vadim V. Tarasov, Helgi B. Schiöth

**Affiliations:** 1Department of Neuroscience, Functional Pharmacology, Uppsala University, 75124 Uppsala, Sweden; 2Advanced Molecular Technology, Limited Liable Company (LLC), 354340 Moscow, Russia

**Keywords:** stem cells, cell therapy, neurological disorders, neurodegenerative disorders

## Abstract

Neurological diseases can significantly reduce the quality and duration of life. Stem cells provide a promising solution, not only due to their regenerative features but also for a variety of other functions, including reducing inflammation and promoting angiogenesis. Although only hematopoietic cells have been approved by the FDA so far, the number of trials continues to expand. We analyzed 492 clinical trials and illustrate the trends in stem cells origins, indications, and phase and status of the clinical trials. The most common neurological disorders treated with stem cells were injuries of brain, spinal cord, and peripheral nerves (14%), stroke (13%), multiple sclerosis (12%), and brain tumors (11%). Mesenchymal stem cells dominated (83%) although the choice of stem cells was highly dependent on the neurological disorder. Of the 492 trials, only two trials have reached phase 4, with most of all other trials being in phases 1 or 2, or transitioning between them (83%). Based on a comparison of the obtained results with similar works and further analysis of the literature, we discuss some of the challenges and future directions of stem cell therapies in the treatment of neurological diseases.

## 1. Introduction

Neurological disorders are heterogenous, and one of the major causes of mortality and general lifespan reduction [1,2]. Causes of neurological conditions vary, ranging from genetically determined to sporadic diseases with unknown etiology, such as neurodegenerative motor neuron disease amyotrophic lateral sclerosis [3]. In most cases, the main aim is to replenish neurons, glia, and blood vessels to achieve positive therapeutic results. Existing drugs are mainly focused on preventing further degeneration, while the successful restoration of structures is still rare [4]. Therapeutic stem cells are a promising candidate for the treatment of neurological conditions with immunomodulatory properties, paracrine effects, and the ability to differentiate into various cell lineages.

Based on the extent of differentiation, stem cells can be divided into totipotent, pluripotent, and multipotent stem cells [5,6,7]. Totipotent cells can form a whole organism and these cells are found in a blastomere that develops from a fertilized ovum. A blastomere gives a further rise to a blastocyst and the inner cell mass of blastomeres represents an example of pluripotent cells, embryonic stem cells (ESCs), that are more committed than the totipotent cells [8,9]. Pluripotent cells can differentiate further to multipotent stem cells, such as umbilical cord-derived mesenchymal stem cells (UC-MSCs), which can only form specific tissue derivatives [10].

Earlier reviews about stem cell treatments (2011) found 123 mesenchymal stem cell (MSC) clinical trials and only 12 of them were focusing on neurodegenerative disorders [11]. A 2016 review on MSCs stated that neurological diseases were the second to bone and cartilage diseases in the number of clinical trials (17.8% and 19.1%), outnumbering even cardiovascular diseases. Amyotrophic lateral sclerosis (12 trials), multiple sclerosis (20), and spinal cord injuries (25) were the three most mentioned groups in clinical trials. Parkinson’s, Alzheimer’s, and Duchenne myopathy had only a handful of clinical trials (3 to 4). In total, there were 200 clinical trials on neurological disorders in year 2016 [12]. A review from 2019 indicates that neurological stem cells (NSCs) were mostly used in stroke (5 trials), while there were only 4 trials each in Parkinson’s disease, lateral sclerosis, multiple sclerosis, and brain tumors. MSCs showed greater numbers in the leading stroke group (27/120) than spinal cord injury (26/120) and multiple sclerosis (23/120). The vast majority of MSCs were said to be from the bone marrow, while adipose-derived stem cells (ADSCs) were in second place and UC-MSCs turned out to be the rarest group among the previous ones. Hematopoietic stem cells were primarily applied only in the cerebral palsy group (17/40) [6]. In 2021, a review was published focusing on MSCs clinical trials starting from 2015. According to their results, neurological disorders and neurological conditions were the second most popular disease treated with MSCs (61 clinical trials) after bone and cartilage diseases [13].

We are observing important technological development leading to new approaches and many new therapies are being tested in the clinic and the number of clinical trials keeps growing. Apart from mesenchymal stem cells, there are many others such as neural and hematopoietic which also are being increasingly taken into consideration. It is getting increasingly challenging to have an overview over the whole field. To our best knowledge, there are no comprehensive quantitative assessments that would systematically cover stem cell clinical trials in the treatment of neurological diseases. Hence, in this review, we present a quantitative analysis of phase, status, stem cell type, and indications of clinical trials with stem cells in neurological diseases. We include clinical trials from 1988 to 2022 to provide a comprehensive overview. We aimed to estimate the general trends and predict future possible directions in comparison to the other reviews. Additionally, in the beginning of this overview, we provide further information about some basics of stem cells research relevant to the development in the field.

## 2. Stem Cell Types

Figure 1 illustrates the different stem cell types and their possible origins, giving preference to stem cells that are related to the treatment of neurological disorders. Bone marrow mononuclear cells (BMMCs) originate from a wider group of bone marrow stem cells BMSC and divide into hematopoietic stem cells and MSCs. Apart from bone marrow, hematopoietic stem cells can be obtained from umbilical cord blood or peripheral blood. From the peripheral blood, stem cells are obtained with the prior mobilization of hematopoietic stem cells from the bone marrow by administrating growth factors. MSCs can be gained from bone marrow, umbilical cord, amniotic fluid, and adipose tissue, with the latter being preferable due to the higher availability of stem cells [14]. Stromal vascular fraction (SVF) can also be attained from adipose tissue. SVF is a specific extract of adipose tissue, apart from mesenchymal stem cells, contains endothelial cells, pericytes, and lymphocytes. Such a greater variety of stem cells, compared to regular ADSCs, could provide a wider range of therapeutic effects. However, there is a lack of standardization due to high variability between different SVFs [15]. The lower part of Figure 1 contains neural stem cells that are located in the subventricular zone and the dentate gyrus of the hippocampus. Besides these zones, neural progenitor cells were also found in the periventricular zone of the spinal cord [16].

## 3. Stem Cell Choice

The choice of stem cells depends on the initial aims of therapy. In the treatment of neurological conditions, the intended results are often cell survival promotion, decreased inflammation, stimulation of neural stem proliferation and angiogenesis. Furthermore, other goals for a stem cell treatment are often stable paracrine effect to support administered stem cells for a long period and minimum side effects. MSCs do amalgamate most of these effects, however, they can differ depending on the source. MSCs have for instance a notable paracrine effect by secreting several growth factors, such as brain-derived neurotrophic factor (BDNF), nerve growth factor, and glial cell line-derived neurotrophic factor (GDNF) that can potentiate regeneration processes. Additionally, ADSCs have a higher rate of proliferation compared to BM-MSCs, as they tend to have higher production of vascular endothelial growth factor (VEGF) and hepatocyte growth factor (HGF). ADSCs were also capable of producing collagen types I, II, and III, while BMMSCs had greater immunosuppressive features which often manifests as the ability to produce anti-inflammatory cytokines. The immunosuppressive properties are crucial when it comes to inflammatory disorders such as multiple sclerosis which results in demyelination of neurons. MSCs have been shown to have anti-inflammatory actions through paracrine activity and cell-to-cell contact with T cells, B cells, natural killer cells, macrophages, monocytes, dendritic cells, and neutrophils. In this way, MSCs induce the production of anti-inflammatory factors, such as IL-10, and by inhibiting the production of pro-inflammatory cytokines IL-12, IFNγ, and TNFα. They can also up-regulate IL-10 and TGF-b production and stimulate surrounding immune cells to further release anti-inflammatory cytokines [26,27]. There is also growing evidence of MSC-produced exosomes which obtain higher immunosuppressive features and can be isolated and used solemnly [28]. On the other hand, umbilical cord mesenchymal stem cells (UC-MSCs) stand out with probably the highest proliferative rate among ADSCs and BM-MSCs [29,30,31].

## 4. Neurological STEM Cells (NSCs)

NSCs are in hard-to-reach areas of the brain which complicates autologous administration. Such cells can also be gained from embryos which, in turn, is complicated with ethical issues [16]. To avoid it, some experimental studies suggest that NSC-like cells could be derived from various other stem cells, such as MSC. In general, NSCs are commonly applied for neuroregeneration as they can differentiate into neurons, astrocytes, and oligodendrocytes, and promote remyelination and axonal growth [16,32,33]. Additionally, NSCs can secrete neuroregeneration-supporting paracrine factors, such as neurotrophic factors, cytokines, and growth factors [16]. Interestingly, NSCs also stimulate angiogenesis, which makes them an attractive therapy in the treatment of stroke [34]. Reportedly, NSCs are potentially able to activate prodrugs by synthesizing specific enzymes (if they carry specific genes) such as cytosine deaminase responsible for converting 5-fluorocytosine into a toxic metabolite 5-fluorouracil. Given NSC’s ability to migrate to the tumor focus, such therapy might appear extremely useful in targeted delivery of chemotherapeutic drugs [33]. Though in multiple sclerosis NSCs clinical efficacy appeared to be questionable, as in multiple cases there were either modest improvement or no significant improvement in the quality of life [35]. Additionally, NSCs, especially those differentiated from induced pluripotent stem cells (iPSCs), contain a certain risk of tumor formation. Although such risk is rarely observed in clinics (possibly due to small sample sizes), it is necessary to consider and apply methods to reduce this carcinogenic effect. One of the possible ways to do this is to reduce the application of transgenic factors of oncogenesis which were found even within the Yamanaka “cocktail” [36].

## 5. Dataset Overview

All trials included in this review were found on one of the largest web resources with information on clinical trials—clinicaltrials.gov. The term “Neurologic Disorder” was used in the field “Condition or Disease” and “Stem Cells” was written in the “Other terms” field. The initial search contained 728 clinical trials. Information about NCT number, title, status, phase, indications, study start year, and the primary completion date was collected for each analyzed clinical trial from clinicaltrials.gov. There were no criteria for inclusion based on gender or age of the participants or trial status. After that, all irrelevant studies were excluded narrowing the data to 492 trials. Irrelevant studies included studies without neurologic disorders, nervous system disorders, or stem cells as an intervention, duplicate studies, and observational studies. All clinical trials that trialed stem cells in the treatment of neurological disorders were included. Stem cells could be used as a monotherapy or in combination with other treatments. Neurological disorders were classified as disease involving brain or peripheral nerve disorders. Every trial was checked manually. All the data collection and filtering were done manually in Excel. The final dataset included only interventional studies starting from 1988 up to 2022. We did not perform a separate search for FDA-approved drugs as it is already provided on their website. This list included only hematopoietic stem cells and it is available by the link—https://www.fda.gov/vaccines-blood-biologics/cellular-gene-therapy-products/approved-cellular-and-gene-therapy-products (accessed on 2 September 2022).

In this work, we review only those cell therapies that have reached clinical trials and no experimental works were included. Classification of stem cells was based solemnly on the data collected. Thus, the dataset included BMSCs which unites BMNCs (bone marrow mononuclear cells), BMMSCs (bone marrow mesenchymal stem cells), BMPCs (bone marrow progenitor cells), UC-MSCs (umbilical cord mesenchymal stem cells), ADSCs, hematopoietic (collected either from bone marrow or from peripheral blood) and also neurological stem cells that amalgamated neural progenitor cells. Based on the column “Condition or disease” on clinicaltrials.gov, we manually isolated various disorders mentioned in clinical trials. We have grouped all brain tumors into one category of “Brain Tumors”. Similarly, a “Stroke” group and a “Various Injuries (brain, spinal cord, peripheral)” group were created. Such groups as encephalopathies, muscular atrophies and dystrophies, inflammatory diseases, leukodystrophies, and eye disorders were also not divided by a specific subtype of disease due to the small number of studies. Disorders such as amyotrophic lateral sclerosis, cerebral palsy, multiple sclerosis, multiple system atrophy, hearing loss, ataxia, Alzheimer, Parkinson, Huntington and Gaucher diseases were named the way they were mentioned in the column “Condition or disease”. The aforementioned information was thenanalysed using MS Excel. There were two clinical trials that reached phase 4, 191 completed studies and 33 trials had results published.

## 6. Results

### Phase and Status

As the first step, all the studies were divided according to their phase and status (Figure 2). Most of the trials were either in the 1st phase (31% and 29% for the trials transitioning from the phase 1 to the phase 2) or in the 2nd phase, which accounted for 23%. The phase 3 contributed to only 3% out of all trials. However, there appeared to be only 2 clinical trials in the phase 4 and thus they were issued in Table 1 separately. According to in Figure 2b, most trials were completed (35%). About 21% of trials were recruiting and only 5% were active.

## 7. Stem Cell Origin

All the trials were divided by the exact stem cell type, as illustrated in Figure 3. Based on the data obtained, we isolated bone marrow stem cells, which were then divided into other subcategories, neural stem cells (including neural progenitor cells), adipose-derived stem cells (or adipose-derived stromal cells), umbilical cord stem cells, hematopoietic, and other mesenchymal stem cells whose origin was not stated or was not clear from the provided information. That notwithstanding, the data contained some stem cells whose origin was notably rare compared to other groups (no more than two clinical trials for each origin of this kind). Consequently, such trials were listed separately.

Based on Figure 3, BMSCs and hematopoietic stem cells were most common in the treatment of neurological disorders in the selected trials (~30%, ~23% accordingly). Within BMSC group, BMMSCs were predominant among others accounting for 51%. The third most frequent group was umbilical cord stem cells standing for 17%. Surprisingly, neural stem cells did not exceed more than 10% out of all the clinical trials. The clinical trial NCT00890032 used autologous brain tumor cells (Figure 3b) and dendritic cells to develop a vaccine for patients with glioblastoma multiforme, which is considered to be one of the most aggressive tumors. NCT04261335 trial is focusing on assessing the safety and tolerability of multi-lineage differentiating stress enduring (MUSE) cells in neonates to further treat ischemic encephalopathy. Dental pulp stem cells (NCT04608838) and liver-derived progenitor cells (NCT01765283) are used for the treatment of acute ischemic stroke and Crigler Najjar syndrome respectively. The latter is also applied for urea cycle disorders. Even though both Crigler Najjar syndrome and urea cycle disorders are not considered to be exactly neurological conditions, their clinical manifestations often include encephalopathy and seizures. Limbal stem cells were primarily isolated for the treatment of ocular injuries, including chemical burns of the cornea (NCT01123044, NCT02202642).

## 8. Indications

All the identified indications in the 492 trials were divided into groups. Attained results are illustrated in Figure 4. As is indicated in Figure 4, most common disorders turned out to be various injuries of the nervous system (14%) including the spinal cord, peripheral nerves, and brain lesions. Stroke, multiple sclerosis, and amyotrophic lateral sclerosis clinical trials also emerged among others accounting for 13%, 12%, and 9% respectively. Implementation of stem cells in patients with brain tumors was as well relatively remarkable and equaled to 11%. Cerebral palsy, Alzheimer’s, and Parkinson’s diseases were more seldomly encountered in comparison to others (not more than 6% for both). The clinical trials focusing on cerebral palsy were just a bit more frequent adding up to 6%. Other indications, such as encephalopathy, epilepsy, leukodystrophy, Duchenne dystrophy (included in muscle atrophy/dystrophy), and diabetic neuropathy were extremely scarce ranging from 1% to a maximum of 3%. Another group that must be mentioned is dedicated to the application of stem cells for the treatment of hearing loss. The small number of studies in this group (1%) can be explained by the fact that this particular direction of using stem cells has begun to be applied relatively recently. A group that is not mentioned in the figure was focusing on “brain death” due to various traumatic brain injuries. We found only one such clinical trial, applying MSCs (NCT02742857).

This figure illustrates all types of diseases mentioned in clinical trials. “Various injuries” consists of “brain injuries” (13 trials), “peripheral injuries” (5 trials), “spinal cord injuries” (54 trials). “Muscle atrophy/dystrophy” consists of “Duchenne muscular dystrophy” (8 trials) and “muscular dystrophy/atrophy” (8 trials). This figure is aimed to illustrate the abundance of different neurological and neurodegenerative conditions that can potentially be treated with stem cells. The most common ones were trials on various injuries (14%), stroke (13%), and multiple sclerosis (12%). Brain tumors were also frequently mentioned (11%), mostly for the reason of “stem cell rescue” to apply higher doses of chemotherapy. The presence of some inherited disorders such as Gaucher’s disease that, apart from other symptoms, can manifest with neurological symptoms opens new perspectives in the treatment of genetic disorders (NCT04145037, NCT00001234, NCT00004294 and NCT01439607).

For the most common disease groups, the specific stem cells used in the trials were illustrated in Figure 5. In most cases, stem cells from the bone marrow predominated. About 22 trials with BMSCs were in the stroke group, 23 were in the multiple sclerosis and the amyotrophic lateral sclerosis groups. The maximum number of BMSCs and ADSCs were used in the group of various injuries, contributing to 33 clinical trials and 15 clinical trials respectively. However, there were no ADSCs applied in both eye disorders and brain tumors. The latter had a significant prevalence of hematopoietic stem cells which were mentioned in 46 clinical trials. The other group which also achieved a notable number of clinical trials with hematopoietic stem cells was multiple sclerosis (17 trials). In all other groups, hematopoietic cells were used extremely infrequently, not exceeding 5 trials in each. Moreover, these cells were completely absent in Alzheimer’s and Parkinson’s groups.

To assess how different types of stem cells are applicable in neurological diseases, we divided studies depending on the origin of stem cells by year. First, we grouped hematopoietic stem cells with BMSC as the most used and mentioned ones (Figure 6). Overall, the use of hematopoietic stem cells seems to be more consistent, starting in 2010 and gradually increasing. However, during a three-year interval from 2018 to 2020, there were only five new clinical trials. At the same time, in 2021, the maximum number of clinical trials with hematopoietic stem cells was registered (more than ten). Another, much smaller peak until 2021 was observed in 2014 and 2015 (both had approximately nine trials). Applying BMSCs for the treatment of neurological disorders and nervous system diseases seems to be more sporadic. For instance, it has risen in 2008 up to 13 trials then dropping to the minimum of 6 trials in the 2009–2011 period. However, it got back in 2012 exceeding 18 clinical trials which was the maximum in the 1988–2022 period. In the last four years (2018–2022), application of BMSCs in nervous system disorders seems to be fading with no more than six clinical trials registered each year. These studies also differ in their start year as hematopoietic stem cells were first registered in 1994 which is 4 years earlier than BMSCs.

BMSCs include BMNCs, MSCs, and bone marrow progenitor cells, while hematopoietic stem cells include those derived from peripheral blood, umbilical cord, or bone marrow. In 2021, there is a dramatic rise in hematopoietic stem cells clinical trials with more than ten trials being registered. The number of hematopoietic stem cell trials is very different in 2022 as only one trial has been registered (with more than 10 registered in 2021). As the year 2022 is not yet finished, we can expect a further rise in the number of trials according to previous years. BMSCs in general exceed larger number of trials, though there seem to be a certain decline in their application in clinics. In 2019, only two trials were registered and in the 2020–2022 the number of trials stayed the same and was equal to 7 (while in, for instance, 2016 and 2017 there were more than 12 trials registered each year).

The same bar charts were constructed for ADSCs and UC-MSCs (Figure 7). Moreover, BMSC in Figure 5 UC-MSCs have no consistent rise or downfall. The period of most uses for UC-MSCs is in 2010–2019 with a maximum of 10 trials in 2011. However, during this period, there was a decrease to 3–4 trials in 2015–2018. In 2019, the number of clinical trials raised to 8 with a notable drop in 2020 to only 3 trials, but a potential to rise as it increases to 5 trials in 2021 and 2022. Application of ADSCs appears to be more confident than UC-MSCs as its period of most uses ranges from 2010 to 2021 with 3 peaks in 2010, 2015, and 2018 (all of them being equal to 5 trials). However, there is a decrease to 2 clinical trials registered in 2022 so it complicates any future prognosis.

Finally, a bar chart for neural stem cells is provided in Figure 8. The largest number of trials was registered in the 2018–2020 period (from 4 to 5 clinical trials). There was also an increase in 2009, 2015, and 2016 (3 trials in all years) and in 2013 there were 4 clinical trials. However, from 2021 to 2022 a fall in NSC application is observed (from 3 to 2 clinical trials). If we compare the total number of studies, then it is neural stem cells that are used the least often. The maximum number of studies for them is 5. The second and third places in terms of rarity of use are divided between UC-MSCs (maximum 10 clinical trials) and ADSCs (maximum value is 5). The most used neural stem cells are hematopoietic cells (more than 10 clinical studies) and BMSCs (more than 18). The achieved results are in accordance with Figure 3. Overall, there is a remarkable decrease in BMSCs application (Figure 5). There might be a possible increase in UC-MSCs, ADSCs, and hematopoietic stem cells (if we do not consider 2022 year as it is not yet finished) (Figure 6 and Figure 7). Any prognosis for neural stem cells is complicated as it has been falling from 2021 (Figure 8).

New neural stem cells clinical trials per year. The bars correspond to the number of new unique clinical trials registered on clinicaltrials.gov for each year. The earliest neural stem cell trial in our dataset was registered in 1997 and thus data start from that year. The number of clinical trials has been increasing since 2007 and reached the peak of five new clinical trials per year in 2019 and 2020. The small number of clinical trials in year 2022 is because the year had not been completed at the time of the writing. There is also a decrease in the number of clinical trials in 2021 compared to 2020. This decrease could be explained by a rise in number of studies focusing on molecular profiles of cells, as well as a detailed study of side effects, which can significantly slow down the onset of clinical trials. However, 2014 witnessed a notable decline in neural stem cells (from 4 clinical trials in 2013 to 1). There were no clinical trials with neural stem cells during the period from 2000 to 2005.

## 9. Discussion

We performed a quantitative analysis of 492 clinical trials focusing on stem cells in neurologic disorders. Using information given on clinicaltrials.gov, we manually isolated the most prominent disorders which were various injuries of brain, spinal cord, and peripheral nerves (14%), stroke (13%), multiple sclerosis (12%), brain tumors (11%), and amyotrophic lateral sclerosis (9%). The most common stem cell treatments were BMSC and hematopoietic stem cells, corresponding to 30% and 23%, although the prevalence was highly dependent on the neurological disorder. Moreover, 83% of all the clinical trials were either in phase 1 or 2, or transitioning between these phases, and only 35% of the clinical trials were completed.

The fast-growing number and diversity of stem cells in clinical trials represent a transformation in the treatment of neurological disorders. In 2019, a review assessing the application of MSCs in clinical trials, found only 27 trials on stroke, 26 on spinal cord injuries, and 23 on multiple sclerosis [6]. In turn, our data illustrated that trials in these groups expanded to 66, 72, and 59 in stroke, various injuries (including spinal cord), and multiple sclerosis groups respectively. It should be mentioned that MSCs appear to be of extreme importance especially in the treatment of stroke as they are the primary source of stem cells in most of trials [37]. This growth is primarily due to stem cells’ ability to promote regeneration, immunomodulation, angiogenesis, and neuroprotection [38,39].

*BMSC & UC-MSC:* Mesenchymal stem cells were the most used stem cell type, accounting for 83% of the clinical trials. 51% of the mesenchymal stem cells were obtained from bone marrow, 17% from the umbilical cord, 11% from adipose tissue, and 4% from unindicated sources. The results are consistent with the existing data [40]. The attractiveness of MSCs is related to their ability to reduce inflammation, by using PGE2 and galectin-1. The immunosuppressive abilities are more pronounced in UC-MSC, which is associated with a defective HLA-II complex blocking further recognition by T-lymphocytes [41]. Furthermore, MSCs are simple to obtain from various sources and can be transplanted in the auto- and allogeneic systems due to their low immunogenicity [42]. Additionally, MSCs can effectively migrate through the transendothelial transition, by using similar receptors as leukocytes (VCAM-1 and TNF-a) [43]. However, the main problem with MSCs lies in inducing the colonization by the transplanted cells due to the blood–brain barrier [42]. Stem cells are usually administered intravenously, but there is evidence that subarachnoid administration would be able to provide the best delivery of stem cells to the damaged area in the CNS [44].

*Hematopoietic stem cells* were prevalently used with brain tumors (46 clinical trials). In the clinical trials, hematopoietic stem cells were commonly combined with radiation therapy and chemotherapeutic drugs, such as etoposide and ifosfamide. Hematopoietic stem cells in these cases were aimed to replenish the damaged bone marrow and allow administration of higher doses of chemotherapy drugs to improve the clinical outcomes in low-grade tumors. Such procedure is often referred to as “hematopoietic stem cell rescue” [45]. However, some authors suggest that applying hematopoietic and neural stem cells may help target cancer stem cells as well. It has been shown that hematopoietic stem cells and neural stem cells can migrate to damaged areas, including areas with cancer stem cells. That migration primarily occurs due to the release of various cytokines by cancer stem cells (osteopontin, cathepsins B, L1, etc.) [46,47,48].

*ADSCs* contributed to 11% of the clinical trials and are expected to grow in the future, as shown in Figure 7. Their efficacy was mostly shown in cardiovascular disorders, due to the great capacity to promote angiogenesis and suppress inflammation [49]. Furthermore, ADSCs are easily accessible through, for example, subcutaneous lipoaspiration. Thus, a rather small number of clinical trials is likely due to the concerns about their ability to cause cancer. Several experimental works showed that ADSCs were able to stimulate the proliferation of breast cancer cells through adipsin release. However, there are works showing that this cancerogenic effect can be weakened, for instance by administrating adipsin-knocked-down ADSCs [50,51].

*NSC:* In turn, according to Figure 3, neural stem cells accounted only for 9% of the stem cell types. Such deficiency in NSCs in clinical trials might be due to ethical issues, as one of the sources for these cells is fetuses and potential risk of tumor formation [16]. Moreover, such transplantation will be considered heterologous which leads to higher risks of autoimmune responses. However, there are now many ways to attain neural stem cells from pluripotent stem cells and by reprogramming somatic cells which should potentiate further application of these stem cells [16,52,53]. There are also many experimental works dedicated to changing NSCs’ profile by overexpressing certain genes, for instance, those that are responsible for the differentiation of neural stem cells into neurons. Therapeutic efficacy in the treatment of stroke can be increased by transfecting the VEGF121 gene into stem cells and, consequently, stimulating angiogenesis [34,54]. From 2018–2021, the number of clinical trials with neural stem cells was remarkably high. As for now, we could expect further development of neural stem cells in the treatment of multiple sclerosis, spinal cord injuries, stroke, and brain tumors.

*Indications:* The most researched indications were the injuries of the brain, spinal cord, and peripheral nerves (14%). The abundance of these clinical trials is likely related to multiple types of cellular damage occurring in the injuries. Stem cells, unlike many other treatments, can provide multiple recovery mechanisms to regenerate neurological networks or ameliorate damage [55]. BMSCs were the most often used stem cell type in this group. The second most researched indication was stroke with high diversity and more than ten different stem cell types. BMSCs were the most common stem cell type also in stroke treatment, but stroke had multiple stem cell types that were absent from all the other indications. One such stem cell type was MUSE-cells which have a low risk of autoimmunity and cancerogenesis compared to many other stem cell types [56,57]. Other popular stem cell indications were multiple sclerosis (12%) and brain tumors (11%). Indications that included surprisingly few clinical trials were, for instance, Parkinson’s disease, Alzheimer’s disease, Encephalopathy and Muscle Atrophy/Dystrophy. Common for these groups is a gap between pre-clinical studies with positive results and clinical trials with the shortage of efficacy [58,59,60,61,62].

## 10. Reported Results

About 33 of the 492 clinical trials reported results and these outcomes were mostly promising but often transient. Especially stem cell treatment of different neuropathies and cancer types were well represented in these clinical trials. In one of the clinical trials, mesenchymal stem cells treatment in diabetic peripheral neuropathy led to an increase of nerve conduction velocity and a decrease of nerve conduction latency. Nerve generation was observed as b-FGF and v-EGF increase from 30.2 pg/mL to 55.4 pg/mL and from 428.7 pg/mL to 601.8 pg/mL respectively after 7 days from the stem cell transfusion (NCT02387749). In another study, hematopoietic stem cell transplantation induced a significant improvement in chronic inflammatory demyelinating polyneuropathy and 80% of the patients achieved immune-medication-free remission 6 months post transplantation (NCT00278629). Various germ cell cancer trials had reported results and one of them showed that a combination of chemotherapy, radiation therapy, and peripheral stem cell transplantation led to therapy response in 74 of 85 patients with intracranial germ cell tumor (NCT00047320). In another study, paclitaxel, ifosfamide, carboplatin, and autologous stem cell transplant were used to treat germ cell tumors that did not respond to cisplatin. In this study, 12 of 23 patients become free of disease for a minimum of 4 weeks and 5 patients showed partial response. Adverse effects included edema, hypokalemia, and dyspnea (NCT00423852). Relapsed germ cell cancer was also treated with stem cell transplantation in another study reporting results. In the study, progression-free survival was 11.8 months after chemotherapy and stem cell transplantation (NCT00002931). Other less common neurological diseases in the clinical trials with reported results were for instance idiopathic inflammatory myopathy and chronic paraplegia. In idiopathic inflammatory myopathy, hematopoietic stem cell transplantation showed efficacy only in three of the seven patients, whereas in chronic paraplegiabone marrow stromal cell decreased chronic pain on IANR-SCIFRS Scale successfully (NCT00278564, NCT01909154).

Many of the clinical trials do not include a control group which is an important weakness. However, there are several studies that have a control group. One such study showed that modified stem cell in the treatment of traumatic brain injury could achieve on average 1.3 units decrease from baseline on 30-point disability rating scale score compared to sham surgery at 24 weeks after treatment. They also observed a slight improvement in global rating of perceived change by the patient and clinicians compared to shame surgery but no significant change in gait velocity (NCT02416492). The clinical trials with reported result had great variability in the outcome but common for many of these clinical trials was that adverse effect hyperglycemia, nausea, and neutropenic fevers were often observed (NCT00787722, NCT00278629, NCT00002931).

## 11. Challenges and Future Directions

One of the major challenges in this field is the low clinical efficacy. As trials also report minimum side effects, there are suggestions to higher the administrated dose. Assessing the minimal dose and the possible dose range is an important task in future to achieve maximum clinical outcomes [40]. In future, there will also likely be seen high grades of stem cell specialization within neurological diseases. Therapeutic goals vary even within a single disease (for example, to lower the inflammation process or stimulate remyelination) and affect the choice of stem cell therapy. Therefore, the specific stem cell choice becomes increasingly important for the exact therapeutical goal. Alongside stem cell origin, other factors, such as route of administration and stem cell preparation are gaining more focus as also these aspects can influence clinical outcomes. Subarachnoid administration of MSCs may for instance promote stem cell migration to CNS and help cross the blood–brain barrier more effectively [38,63]. Furthermore, while autologous transplantation is often the preferable stem cell source, it is not always possible with certain types of stem cells. This has been increasingly solved with allo-stem cells that have the advantage with regard to time, quality assurance, and cost [64].

## 12. Conclusions

Neurological disorders, being extremely diverse and complex, remain a global burden for both patients and clinicians in the search for appropriate therapy. Stem cells can be an excellent solution given their ability to stimulate tissue regeneration, which is most important in the recovery and treatment of neurological conditions. However, cell therapy remains to be a highly unexplored area with many undiscovered mechanisms of action and other features of cells. The regenerative potential of stem cells may not be the only target for study. It has been shown that such properties as reducing inflammation or activating prodrugs (concerning neural stem cells) may be more important and necessary in the context of a particular disease. However, the lack of theoretical knowledge is not the only problem cell therapy might be facing. There is a fairly noticeable gap between preclinical studies and actual clinical trials. It is common that results that were obtained at the preclinical stages do not correspond to those achieved in clinics. The lack of standardized criteria for evaluating the clinical efficacy of stem cells also complicates any trials. However, despite all these difficulties, the integration of stem cells into clinics is becoming more confident, as shown by the 492 clinical trials found in this review. Neural stem cell, ADSC, and UC-MSC clinical trials have earlier been outnumbered by the clinical trials with BMSC and hematopoietic stem cells. Despite the slight decrease in the number of clinical trials in 2020 and 2021, we expect these stem cell groups to become increasingly popular as stem cell treatments become more specialized and disease targeted. Due to the high heterogeneity of diseases, we also expect an increase in various beneficial combination therapies uniting standard drugs and cell therapy.

## Figures and Tables

**Figure 1 ijms-23-11453-f001:**
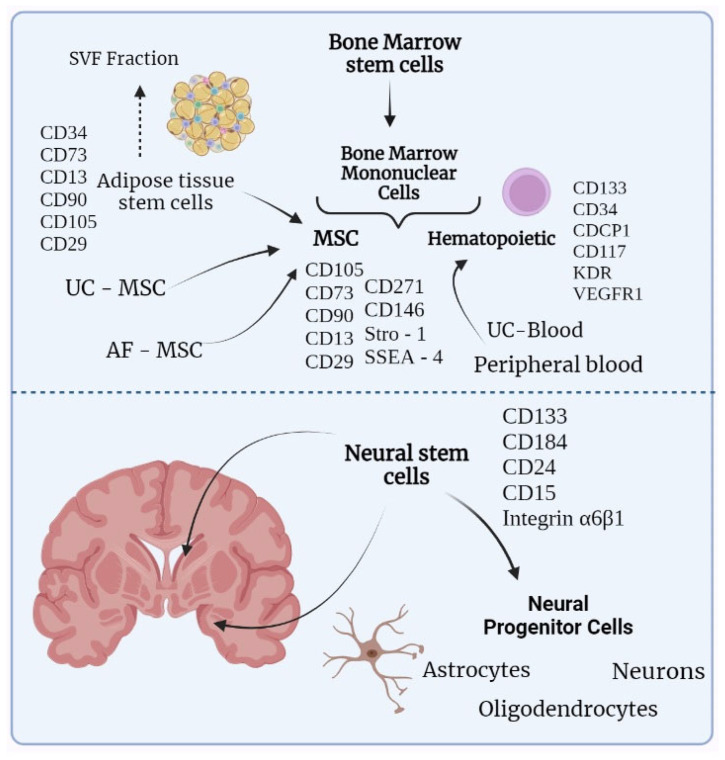
Various types of stem cells and their possible origins. This figure is presented for a general understanding of the possible sources of stem cells and is therefore greatly simplified. More detailed classifications can be found in the following papers [17,18,19,20]. Additionally, markers were indicated for some stem cells, the presence of which makes it possible to isolate certain cells. It is worth noting that the classification by surface markers includes more criteria, including the absence of certain markers. In turn, surface proteins can vary significantly. For example, in UC-MSCs there is no CD271, but there is CD146, which is also present on mesenchymal cells of the bone marrow and adipose tissue. *Stro-1*, like *SSEA-4*, was found only in MSCs from the bone marrow. For a more detailed description of the features of surface markers, we propose the following works [21,22,23,24,25]. In general, mesenchymal stem cells can be derived from a variety of tissues including bone marrow, adipose tissue, amniotic fluid, and umbilical cord. While all cells from these origins would be considered as mesenchymal, they would still differ in some properties such as immunosuppressive function. Hematopoietic stem cells are found in bone marrow, umbilical cord blood, and also in peripheral blood. The lower part of the figure contains an illustration of neural stem cells, originally deriving from the subventricular zone and the dentate gyrus of hippocampus. Neural stem cells are able to differentiate into oligodendrocytes, astrocytes, and neurons. Arrows are indicating the exact parts of the brain where neural stem cells are thought to be located. AF-MSC—amniotic fluid mesenchymal stem cells, UC-MSC—umbilical cord mesenchymal stem cells, SVF fraction—stromal vascular fraction, MSC—mesenchymal stem cells, UC-Blood—umbilical cord blood, VEGFR2 - vascular endothelial growth factor receptor 2.

**Figure 2 ijms-23-11453-f002:**
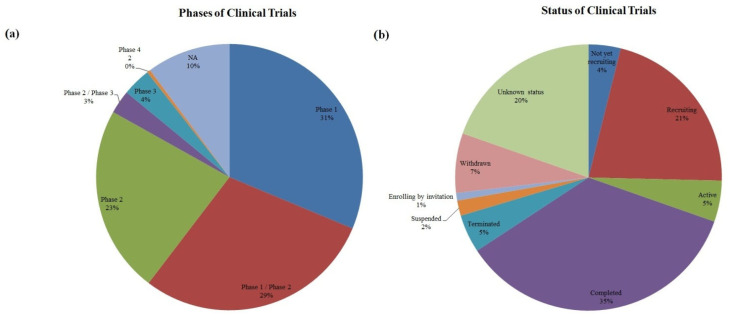
This figure provides a graphical representation of phases and statuses in 492 selected trials. The majority of trials were completed (35%), 21% of trials were recruiting while only 5% were active. We also detected a high percentage of trials with unknown status—20%. 7% of all trials were withdrawn. Phase 1 and phase 2 trials were predominant accounting for 83%. Only 4% of trials were in phase 3 and only 2 trials reached phase 4. (**a**) Represents phases of the clinical trials and (**b**) Represents statuses of the clinical trials, NA—not applicable.

**Figure 3 ijms-23-11453-f003:**
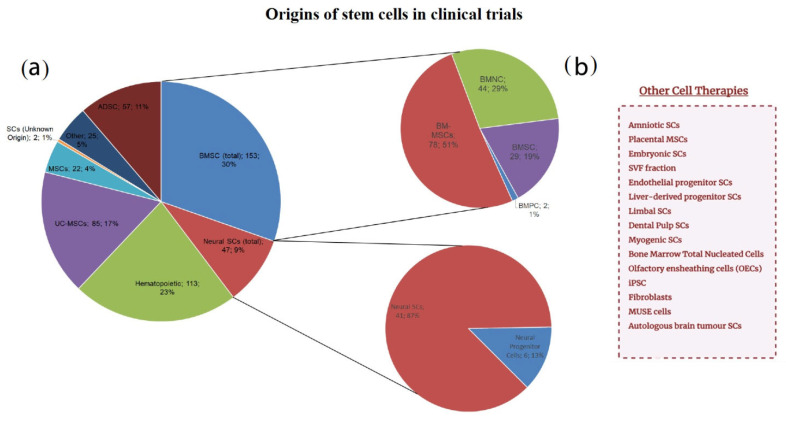
This figure contains information about various stem cell origins mentioned in the clinical trials. Stem cells derived from bone marrow (BMSC) were the largest group accounting for 30%. Within this group 51% of trials applied mesenchymal stem cells (BM-MSC). Hematopoietic stem cells were the second largest group being equal to 23% of all trials. Neural stem cells were present in trials in only 9% of cases losing to ADSCs (11%) and UC-MSCs (17%). It is worth noting the high diversity of different stem cells for the treatment of neurological diseases including dental pulp stem cells, MUSE cells and autologous brain tumor stem cells. Induced pluripotent stem cells (iPSCs), which are widely present among experimental studies, were found in only one clinical trial which was terminated for the reason of changes in research objectives (NCT02246491). (**a**) Represents various types of stem cells derived from different origins. In some cases, it was not clear where exactly mesenchymal stem cells were derived from, so such trials were marked as MSC. In cases in which clinical trials only mentioned “stem cells” without specifying the exact origin, they were marked as SCs of unknown origin. Neural stem cells and neural progenitor cells were grouped. The same was applied to all the stem cells derived from bone marrow. “Other” group included all the uncommon stem cell origins which were then provided independently. Neural SCs—neural stem cells, UC-MSCs—umbilical cord mesenchymal stem cells, ADSC—adipose tissue-derived stem cells, BMSC—bone marrow stem cells, BMNC—bone marrow mononuclear cells, BMPC—bone marrow progenitor cells, BM-MSCs—bone marrow mesenchymal stem cells, SCs—stem cells, MSCs—mesenchymal stem cells, (**b**) lists all rare stem cells origins (those that were included in no more than two clinical trials), iPSC—induced pluripotent stem cells, MUSE—multi-lineage differentiating stress enduring cell.

**Figure 4 ijms-23-11453-f004:**
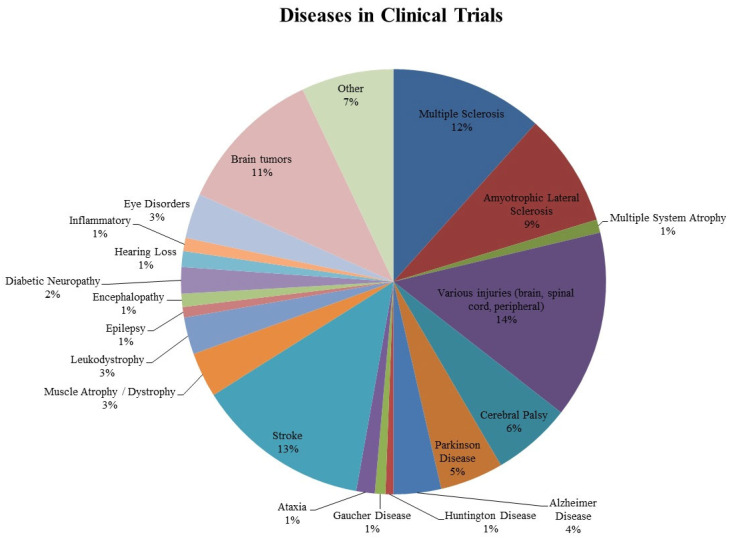
Conditions in clinical trials.

**Figure 5 ijms-23-11453-f005:**
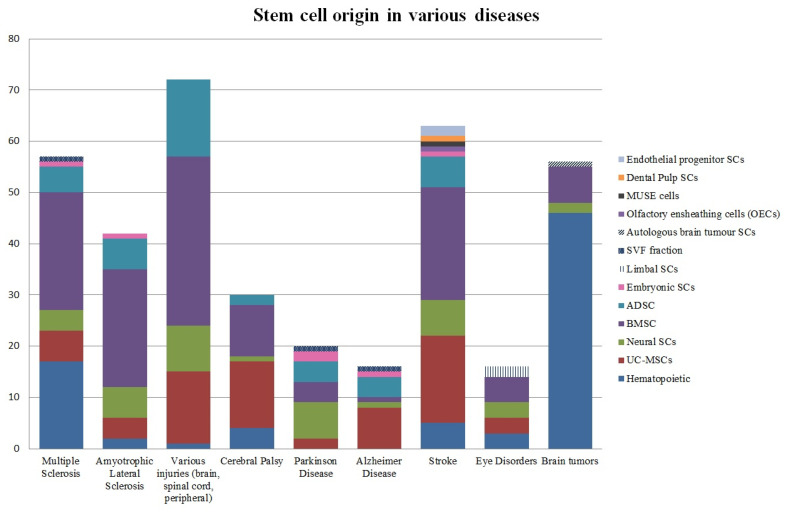
Types of stem cells in different conditions. Olfactory ensheathing cells (OEC’s) (NCT01327768), MUSE cells (NCT04261335), endothelial progenitors (NCT01468064 and NCT02605707), dental pulp SC’s (NCT04608838) were only encountered in “stroke” group making it the most diverse group among others. Limbal SC’s were only found in “eye disorders” group. SVF fraction was present in “Parkinson’s” (NCT03297177), “Alzheimer’s” (NCT03297177), and “multiple sclerosis” (NCT04849065) groups. There were two clinical trials with limbal *stem* cells in eye disorders group. Both of these trials were dealing with eye injuries (NCT02202642 and NCT02290886). Autologous brain tumor cells were mentioned only in “Brain Tumors” group concerning the development of vaccine therapy for the recurrent glioblastoma (NCT00890032). In turn, Alzheimer’s group was the only group where the neural stem cells were the most used stem cells. Interestingly, the Parkinson’s group turned out to have only one clinical trial with neural stem cells. It is the UC-MSCs that were better established in this disease group (8 clinical trials). Cerebral Palsy was a group with the greatest number of trials with UC-MSCs (13 trials). These stem cells also seemed to play a huge role in stroke treatment (17 trials). The use of scarcer encountered types of stem cells largely depended on the specific group of diseases. So, limbal stem cells were only represented in the group of eye diseases and endothelial progenitor cells could only be found in the stroke group. That notwithstanding, SVF fraction was found in Alzheimer’s, Parkinson’s, and multiple sclerosis groups. The group with the most diversity in the stem cell types turned out to be the stroke group, with ten different stem cell types, including dental pulp stem cells, embryonic stem cells, and MUSE cells. ESCs appeared to be widely distributed among the groups (present in amyotrophic lateral sclerosis, multiple sclerosis, Parkinson’s, Alzheimer’s, and stroke disease groups) although such cells did not exceed large numbers as there were usually no more than two clinical trials on ESCs.

**Figure 6 ijms-23-11453-f006:**
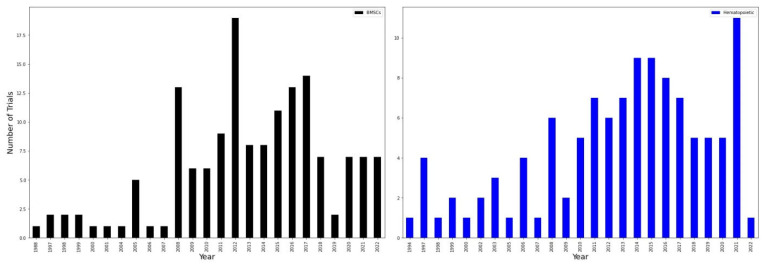
BMSCs and hematopoietic stem cells clinical trials. On the **left** (black) is a graph for BMSCs and on the **right** (blue) for hematopoietic stem cells.

**Figure 7 ijms-23-11453-f007:**
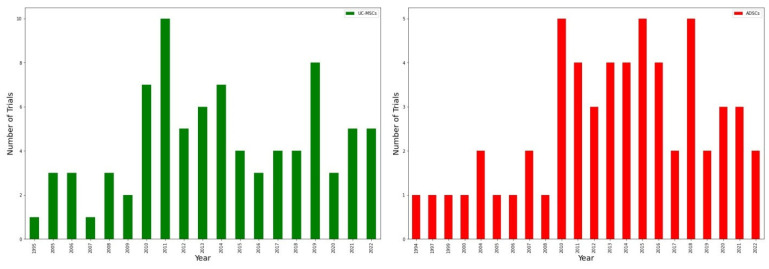
ADSCs and UC-MSCs clinical trials. On the **left** (green) is a graph for UC-MSCs and on the **right** (red) for ADSCs. Both are a source for mesenchymal stem cells and have generally similar graphs, and, therefore, approximately the same popularity in the clinic. On the other hand, in 2016 and 2020 there is a drop to three clinical trials for UC-MSCs, while for ADSC’s such drop was registered in 2017 and 2019 (two trials in each year). Overall, UC-MSCs seem to be more frequently applied with the highest peak in 2011 when 10 trials were registered. In turn, ADSCs had three major peaks in 2010 (5 trials), 2015 (5 trials), and 2018 (5 trials).

**Figure 8 ijms-23-11453-f008:**
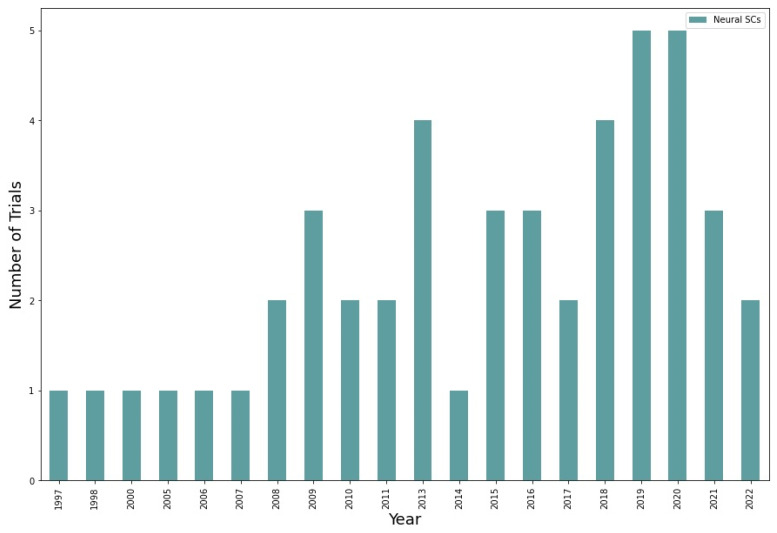
Neural stem cells clinical trials.

**Table 1 ijms-23-11453-t001:** Clinical trials that have reached phase 4.

NCT Number	Title	Status	Condition	Interventions
NCT03633565	Comparative Study of Strategies for Management of Duchenne Myopathy (DM)	Unknown status/No Results Available	Duchenne Myopathy	Sildenafil Prednisolone **Mesenchymal** stem cell transplantation
NCT00336531	Efficacy of Prophylactic Itraconazole in High-Dose Chemotherapy and Autologous Hematopoietic Stem Cell Transplantation	Completed/No Results Available	Neuroblastoma Brain Tumor Retinoblastoma Wilms Tumor Mycoses	Itraconazole Autologous **Hematopoietic** Stem Cell Transplantation

## Data Availability

All data tables are available at the authors upon request.

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
