# Peer review of "Stem Cells in Clinical Trials on Neurological Disorders: Trends in Stem Cells Origins, Indications, and Status of the Clinical Trials"

_ijms, 2022, doi:10.3390/ijms231911453_

Round 1
Reviewer 1 Report
This review is concerning on stem cells therapies in neurological disorders and include clinical trials from 1988 to 2022. The authors present an analysis of phase and status of clinical trials, stem cell type, and indications in the 492 clinical trials for stem cells therapy in neurological diseases.
Major:
On the page 5, line 184, the Authors state that “…….33 trials had results published”. In the results / discussion, I would expect a summary of the published results of these clinical studies, what kind of diseases were involved, what was the success rate if any.
Minor:
The term “Figure” is writing in various style: “Figure, figure, fig, Fig “ should be corrected accordingly to IJMS requirement in whole manuscript.
Also References are given in a different style, they should be unified according to the IJMS guidelines
Author Response
We thank the reviewer for the thorough comments on our manuscript; below you can find all the answers to your comments
- On the page 5, line 184, the Authors state that “…….33 trials had results published”. In the results / discussion, I would expect a summary of the published results of these clinical studies, what kind of diseases were involved, what was the success rate if any.
We have elaborated description of clinical trials in the Discussion part of our manuscript, which appear as follows:
“Reported results: 33 of the 492 clinical trials reported results and these outcomes were mostly promising but often transient. Especially stem cell treatment of different neuropathies and cancer types were well represented in these clinical trials. In one of the clinical trials, mesenchymal stem cells treatment in diabetic peripheral neuropathy led to an increase of nerve conduction velocity and a decrease of nerve conduction la-tency. Nerve generation was observed as b-FGF and v-EGF increase from 30.2pg/ml to 55.4pg/ml and from 428.7pg/ml to 601.8pg/ml respectively after 7 days from the stem cell transfusion (NCT02387749). In another study, hematopeotic stem cell transplantation induced a significant improvement in chronic inflammatory demyelinating polyneu-ropathy and 80% of the patients achieved immune-medication free remission 6 months post transplantation (NCT00278629). Various germ cell cancer trials had reported results and one of them showed that a combination of chemotherapy, radiation therapy and peripheral stem cell transplantation lead to therapy response in 74 of 85 patients with intracranial germ cell tumor (NCT00047320). In another study, paclitaxel, ifosfamide, carboplatin and autologous stem cell transplant were used to treat germ cell tumors that did not respond to cisplatin. In this study 12 of 23 patients become free of disease for a minimum of 4 weeks and 5 patients showed partial response. Adverse effects included edema, hypokalemia, and dyspnea (NCT00423852). Relapsed germ cell cancer was also treated with stem cell transplantation in another study reporting results. In the study progression free survival was 11.8 months after chemotherapy and stem cell trans-plantation (NCT00002931). Other less common neurological diseases in the clinical trials with reported results were for instance idiopathic inflammatory myopathy and chronic paraplegia. In idiopathic inflammatory myopathy hematopoetic stem cell transplanta-tion showed efficacy only in 3 of the 7 patients, whereas and in chronic paraplegiabone marrow stromal cell decreased chronic pain on IANR-SCIFRS Scale successfully (NCT00278564, NCT01909154).
Many of the clinical trials do not include a control group which is an important weakness. However, there are several studies that have a control group. One such study showed that modified stem cell in the treatment of traumatic brain injury could achieve on average 1.3 units decrease from baseline on 30-point disability rating scale score compared to sham surgery at 24 weeks after treatment. They also observed a slight improvement on global rating of perceived change by the patient and clinicians com-pared to shame surgery but no significant change in gait velocity. (NCT02416492) The clinical trials with reported result had great variability in the outcome but common for many of these clinical trials was that adverse effect hyperglycemia, nausea, and neu-tropenic fevers were often observed (NCT00787722, NCT00278629, NCT00002931).”
- The term “Figure” is writing in various style: “Figure, figure, fig, Fig “ should be corrected accordingly to IJMS requirement in whole manuscript.
It was corrected.
- Also References are given in a different style, they should be unified according to the IJMS guidelines
All the references are now modified according to IJMS guidelines.
Reviewer 2 Report
Reviewer comments and suggestions
The study highlighted the importance of stem cells in preventing neurological diseases. For the study hypothesis, the authors analyzed 492 clinical trials and illustrate the trends in stem cell origins, indications and phase and status of the clinical trials, to provide a better understanding and predictions of future trends. The study suggested that the most common neurological disorders treated with stem cells were injuries of brain, spinal cord, and peripheral nerves (14%), stroke (13%), multiple sclerosis (12%), and brain tumors (11%). Mesenchymal stem cells dominated (83%) although the choice of stem cells was highly dependent on the neurological disorder. Based on this review the authors concluded some of the challenges and future directions of stem cell therapies in the treatment of neurological diseases.
Overall, the study provides a good insight into the stem cells for use in neurological disorders to protect from ageing-related diseases. However, the authors did not mention the analysis or the methods used in the study, which was lacking. I could not find how they selected or analyzed a total of 492 studies.
A few comments are below to be incorporated into the manuscript.
- Line 31, please mention a few examples otherwise the authors have to explain ALS
- Line 135-136 need to explore
- Line 150-152 Sentences need to be clear for the common reader of your paper
- Figure 3,4 what was the source and methods to point out the % of disorders
- Figure 8 Why the trails have been reduced in recent years, please explain the reason
- Discussion Which software was used or manually you did. there should be some parameters/methods to analyze
- Line 511 Your data shows decreasing
- All references should be modified based on the MDPI journals.
Author Response
We appreciate all the effort put into reviewing our manuscript and helping to improve it; below you can find all the answers to your comments
- The authors did not mention the analysis or the methods used in the study, which was lacking. I could not find how they selected or analyzed a total of 492 studies.
We added a more detailed description of the selection and analysis of trials in the Dataset overview section of the article. Newly added information can be found below:
“All clinical trials that trialed stem cells in the treatment of neurological disorders were included. Stem cells could be used as a monotherapy or in combination with other treatments. Neurological disorders were classified as diseases involving brain, spinal cord or peripheral nerve disorders. Every trial was checked manually. All the data collection and filtering were done manually in Excel.”
- Line 31, please mention a few examples otherwise the authors have to explain ALS
ALS in line 31 was specified.
- Line 135-136 need to explore
We further explored lines 135-136, which appear as follows:
“MSCs have been shown to have anti-inflammatory actions through paracrine activity and cell-to-cell contact with T cells, B cells, natural killer cells, macrophages, monocytes, dendritic cells, and neutrophils. In this way, MSCs induce the production of anti-inflammatory factors, such as IL-10, and inhibit the production of pro-inflammatory cytokines IL-12, IFNγ, and TNFα.”
- Line 150-152 Sentences need to be clear for the common reader of your paper
We changed and explored information in lines 150-152 to add a more detailed description. The newly added text is given below:
“In general, NSCs are commonly applied for neuroregeneration as they can differentiate into neurons, astrocytes, and oligodendrocytes, and promote remyelination and axonal growth [16, 23, 24]. Additionally, NSCs can secrete neuroregeneration-supporting paracrine factors, such as neurotrophic factors, cytokines, and growth factors [16]. Interestingly, NSCs also stimulate angiogenesis, which makes them an attractive therapy in the treatment of stroke [25]. Reportedly, NSCs are potentially able to activate prodrugs by synthesizing specific enzymes (if they carry specific genes) such as cytosine deaminase responsible for converting 5-Fluorocytosine into a toxic metabolite 5-Fluorouracil. Given NSC's ability to migrate to the tumor focus, such therapy might appear extremely useful in the targeted delivery of chemotherapeutic drugs [24].”
- Figure 3,4 what was the source and methods to point out the % of disorders
The exact process of counting the percentage of diseases was added in the Dataset overview part of the manuscript, which appears as follows:
“Based on the column “Condition or disease” on clinicaltrials.gov, we manually isolated various disorders mentioned in clinical trials. We have grouped all brain tumors into one category of "Brain Tumors". Similarly, a “Stroke” group and a “Various Injuries (brain, spinal cord, peripheral)” group were created. Such groups as encephalopathies, muscular atrophies and dystrophies, inflammatory diseases, leukodystrophies and eye disorders were also not divided by a specific subtype of disease due to the small number of studies. Disorders like amyotrophic lateral sclerosis, cerebral palsy, multiple sclerosis, multiple system atrophy, hearing loss, ataxia, Alzheimer’s, Parkinson’s, Huntington’s and Gaucher’s diseases were named the way they were mentioned in the column “Condition or disease”. The aforementioned information was then used to create figures 3, 4, and 5 using MS Excel for quantitative analysis.”
- Figure 8 Why the trails have been reduced in recent years, please explain the reason
We added one of the most possible reasons for the decline in the number of clinical trials with Neural Stem Cells in the figure 8 description. The added text appears in the manuscript as follows:
“The small number of clinical trials in year 2022 is because the year had not been completed at the time of the writing. There is also a decrease in the number of clinical trials in 2021 compared to 2020. This decrease could be explained by a rise in number of studies focusing on molecular profiles of cells, as well as a detailed study of side effects, which can significantly slow down the onset of clinical trials.”
- Discussion Which software was used or manually you did. there should be some parameters/methods to analyze
We did a manual analysis of all the trials which is elaborated in the Dataset overview and we also added information in the Discussion part, which appears as follows:
“Using information given on clinicaltrials.gov, we manually isolated the most prominent disorders which were various injuries of brain, spinal cord, and peripheral nerves (14%), stroke (13%), multiple sclerosis (12%), brain tumors (11%), and amyotrophic lateral sclerosis (9%).”
- Line 511 Your data shows decreasing
Despite the actual decrease in neural stem cell application, we think that these cells still have a huge potential in the future. With the further development of molecular technologies, we feel that there is a possibility to minimize the current problems with these stem cells. We added the possible reason for an increase in neural stem cells in the Conclusion part, acknowledging that at this moment data illustrates a decrease in neural stem cells. Added text appears as follows:
“Neural stem cell, ADSC, and UC-MSC clinical trials have earlier been outnumbered by the clinical trials with BMSC and hematopoietic stem cells. Despite the slight decrease in the number of clinical trials in 2020 and 2021, we expect these stem cell groups to become increasingly popular as stem cell treatments become more specialized and disease targeted.”
- All references should be modified based on the MDPI journals.
All the references have been changed according to MDPI style.
Round 2
Reviewer 1 Report
The authors responded to my comments sufficiently improving the quality of the manuscript. The paper can be publish in present form.